# Gender Disparities in Osteoporosis Knowledge, Health Beliefs and Preventive Behaviors in Najran City, Saudi Arabia

**DOI:** 10.3390/nu15163658

**Published:** 2023-08-21

**Authors:** Heba A. Ibrahim, Mohammed H. Nahari, Mugahed A. Al-khadher, Nemat I. Ismail, Wafaa T. Elgzar

**Affiliations:** 1Department of Maternity and Childhood Nursing, Nursing College, Najran University, Najran 66441, Saudi Arabia; heaibrahim@nu.edu.sa; 2Department of Clinical Laboratory Sciences, College of Applied Medical Sciences, Najran University, Najran 66441, Saudi Arabia; mhnahari@nu.edu.sa; 3Department of Medical Surgical Nursing, Nursing College, Najran University, Najran 66441, Saudi Arabia; maalkhadher@nu.edu.sa; 4Department of Obstetrics and Gynecology Nursing, Nursing College, Damanhour University, Damanhour 22511, Egypt; nematt_ismail@nur.dmu.edu.eg

**Keywords:** osteoporosis, health behaviors, gender, knowledge, health beliefs

## Abstract

Osteoporosis is a chronic bone disease affecting both men and women, but it is more prevalent in women. Promoting a healthy lifestyle among adults, particularly women, is crucial in preventing and reducing the osteoporosis impact. This study aimed to compare the osteoporosis knowledge, health beliefs and preventive behaviors among adult male and female in Najran city Saudi Arabia. This cross-sectional study was performed on 516 males and 581 females in Najran City, Saudi Arabia, from January to April 2023. The data collection instrument is a self-reported online questionnaire consisting of basic data, the Osteoporosis Preventive Behavior (OPB) scale, the osteoporosis knowledge assessment tool, and the osteoporosis health belief scale. The results revealed that male participants had a higher OPB (26.70) than females (20.32). However, females have a higher knowledge (10.71), perceive themselves as more susceptible to osteoporosis (20.34) and had higher exercise barriers (20.11) compared to males (9.97, 18.79 and 19.20, respectively). Statistically significant correlations (*p* < 0.001) were observed between OPB, osteoporosis knowledge (r = 0.26), perceived susceptibility (r = 0.33), severity (r = 0.53), exercise (r = 0.54) and calcium (r = 0.33) benefits, exercise (r = 0.40) and calcium (r = 0.81) barriers and health motivation (r = 0.37). The study concluded significant disparities between males and females regarding osteoporosis-related knowledge, preventive behaviors, perceived susceptibility, severity, and exercise perceived barriers. The results suggest gender-based educational interventions to enhance OPB by addressing osteoporosis-related knowledge, perceived susceptibility, seriousness, benefits, and health motivation.

## 1. Introduction

Osteoporosis is a chronic bone disease that affects both genders, but it is more prevalent in females. It is most commonly seen among women after menopause and older men [1]. Osteoporosis risk is increased in diabetic and obese patients; both of these diseases are very common in Saudi Arabia [2,3]. Osteoporosis is characterized by a gradual bone density reduction, which makes bones more fragile and increases the likelihood of fractures [4]. Osteoporosis-related fractures negatively affect one’s quality of life, resulting in pain, limited movement, deformity, and depressive mood. In addition, the high-mortality rate following a hip or spinal fracture due to osteoporosis is common among the older population [5].

Osteoporosis prevalence varies by country and region. A recent meta-analysis reported that osteoporosis is prevalent among 18.3% of the global population, with the highest prevalence observed in Africa [6]. Several Saudi studies explored osteoporosis prevalence among adults, with an estimated range from 11.8% to 39.5%. It is worth mentioning that some of these research were conducted over 15 years ago [7,8,9,10]. The impact of osteoporosis on disability is higher than any individual cancer, excluding lung cancer. It is also equivalent to or higher than the disability caused by chronic diseases like rheumatic arthritis, bronchial asthma, and hypertensive-related cardiac disorders [11,12]. According to previous studies, hip fractures in Saudi Arabia have a 1-year mortality rate of 27%, which is higher than the average rate of 22%, according to the systematic review involving 36 countries from six continents [13,14]. The hip fracture-related cost in Saudi Arabia is anticipated to reach USD 9.34 billion by 2025, due to rapid population growth and a longer life span. Despite this significant burden, osteoporosis is often not diagnosed or treated properly, and many patients are unaware that they have osteoporosis before experiencing a fracture. Therefore, it is important to identify the osteoporosis risk factors to identify high-risk groups [15].

Osteoporosis is influenced by two types of factors: modifiable and non-modifiable factors. Modifiable lifestyle factors include poor dietary habits, smoking and Alcohol, and sedentary life, while non-modifiable factors include age, genetics, and race. It is important to increase bone density during childhood and adolescence, maintain bone mass during the adult period, and minimize bone loss in older age to prevent or reduce the severity of osteoporosis [16,17]. While osteoporosis can impact both genders, it is more prevalent in women, with 80% of cases affecting them. Although both males and females experience a gradual loss of bone density, females tend to experience it earlier and at a faster rate. However, men suffer higher mortality rates after low-traumatic fracture related-osteoporosis of the hip, femur, and pelvis [18,19]. Therefore, promoting a healthy lifestyle among adults, particularly women, is crucial in preventing and reducing the impact of osteoporosis. It is recommended to engage in weight-bearing activities, maintain a healthy body weight, consume enough calcium, refrain from smoking, and limit Alcohol, coffee, and sodium intake from an early age to promote bone health [20]. In accordance, high-risk groups need educational interventions to increase public awareness about healthy lifestyles. Educational interventions are assumed to be more effective when it is needs-based and built on an educational model.

Health educational models are useful for researchers in identifying the factors that influence behaviors. These models help in understanding the impact of these factors on behaviors and address personal motivations for preventive behavior [21]. Among these models is the Health Belief Model (HBM), developed by Rosenstock and widely used to explain why persons utilize preventative health services. The HBM suggests people act based on perceived risks and benefits [22]. It is commonly used to understand why people engage in preventative health behaviors such as engaging in a healthy lifestyle, including dietary modifications, physical activity, and regular health checks. The HBM is frequently applied to disorders like infectious diseases, malignancy, and osteoporosis. People are most likely to engage in preventative behaviors when they feel vulnerable to a health condition, believe it is serious, see the benefits of taking action as greater than the barriers, and are motivated to be healthy [22,23]. In order to develop effective strategies for optimizing peak bone mass in young adults, it is crucial to have an insight and understanding of their knowledge, beliefs, and practices of preventive behavior related to osteoporosis [24]. Therefore, the present study aimed to explore the gender disparities in osteoporosis-related knowledge, health beliefs, and preventive behaviors in Najran City, Saudi Arabia.

## 2. Subjects and Methods

### 2.1. Design and Setting

A comparative cross-sectional study was carried out in Najran city, located on the southern border of Saudi Arabia with Yemen. The city has a population of 184,772 individuals aged 20 years and above, comprising various nationalities. It is also the capital of the Najran region [25].

The sample size was computed using the free Epi-info online program based on population size = 184,772; the anticipated frequency of the population having high OPB is 50%, absolute precision is 5%, and design effect is 1%. The calculated sample size is built on a 99.9% confidence interval. The sample size is 1077 male and female participants; after adding 5% for anticipated unreliable data, the sample size was 1131.

Initially, 1131 participants were recruited then 34 sheets were excluded due to inconvenient data and the final data analysis conducted on 1097 sheets. The participants were recruited using a convenience sampling technique from Najran City using an online survey. The inclusion criteria include early and middle adult males and females aged 20–65 years, and more who can read and write and agree to contribute to the survey.

### 2.2. Data Collection Instruments

The data collection tool is a structured self-reported questionnaire composed of four parts.

Part 1: Participants’ demographic characteristics and osteoporosis history. It includes age, Body Mass Index (BMI), residence, occupation, monthly income, education, marital status, personal and family history of osteoporosis, personal history of bone fracture, and attendance of any osteoporosis educational interventions.

Part 2: Osteoporosis preventive behavior scale (OPBS): The researchers reviewed the related literature and then developed this scale to asses OPB [26,27]. OPBS incorporates seven statements rated on a 5-point Likert scale ranging from always (5), usually (4), sometimes (3), rarely (2), and never (1). The total scale score ranged from 7–35, with the higher score indicating an OPB. Low OPB was considered when the scale score ranged from 7–21, and high OPB was considered at a scale score of 22–35. The OPBS was validated in the current study by a jury of four experts and a biostatistician and the Cronbach alpha coefficient showed good reliability in current study for the OPB scale (r = 0.821).

Part 3: Osteoporosis Knowledge Assessment Tool (OKAT) [28], Winzenberg et al., 2003, developed it to evaluate participants’ osteoporosis-related knowledge. The scale comprises 20 true or false questions that evaluate four basic domains; osteoporosis symptoms, risk factors; preventive measures such as physical exercises, healthy diet; and treatment opportunities. For coding, the right answer scored “1”, and the wrong answer scored “0” The overall scale score ranges from 0 to 20, with a higher score demonstrating higher knowledge regarding osteoporosis. The total score of the participant was used to determine whether their knowledge was unsatisfactory (0–12) or satisfactory (13–20). The Arabic version of OKAT had good internal consistency (r = 0.824) [29].

Part 4: Osteoporosis Health Beliefs Scale (OHBS) [30]. Kim et al., 1991, developed it to evaluate participants’ osteoporosis-related health beliefs. The scale encompasses osteoporosis susceptibility (6 statements), seriousness (6 statements), exercises and calcium intake benefits (12 statements), exercising and calcium intake barriers (12 statements), and health motivation (6 statements). The OHBS is rated on a 5-point Likert scale, ranging from strongly disagree (1) to strongly agree (5). The total score for each subscale is obtained by summing items with higher scores, thus indicating higher beliefs. The OHBS Arabic version had high reliability according to Sayed–Hassan and Bashour (r = 0.806) [29]. The data collection tools were tested for face validity by a jury of 5 experts in the field and a biostatistician.

Data collection started from January to April 2023. A structured online questionnaire was used to collect data through Google Forms. The authors created a one-page recruitment sheet shared with individuals and groups through social media, e.g., WhatsApp, Telegram, Facebook, Instagram, Twitter, and email. The sheet provided information about the study’s purpose, procedures, voluntary participation, confidentiality, and instructions for completing the questionnaire. It also included a link to the online questionnaire. All the online questionnaire items are made as required to ensure data completeness.

Ethical approval: After the agreement of the deanship of scientific research at Najran University on the research proposal, another ethical approval was taken from the ethical committee of Najran health affairs (IRB NO 2023-14 E). The researchers took informed consent from each participant after an explanation of the study’s purpose and participants’ anonymity was applied. The information collected was solely utilized for research, and individuals were made aware that they had the option to opt out of participation without any consequences. The informed consent was written at the beginning of the electronic questionnaire, and the agreement was essential to proceed.

Data analysis: IBM version 22 was used to analyze data. In the beginning, a data check was performed, and 34 sheets were excluded because of inconvenient data, and the data analysis was conducted on 1097 sheets. All the study variables were compared between men and women, and the descriptive statistics were executed to represent descriptive data. Among the study variables, residence, occupation, monthly income, history of osteoporosis, and bone fracture were nominal. Osteoporosis knowledge, preventive behaviors, and health beliefs were numerical variables. The total osteoporosis knowledge, preventive behaviors, and health beliefs were obtained by total items. The disparities among groups were checked using chi-square, fisher exact, and *t*-test. The associations between OPB and osteoporosis knowledge and health beliefs were examined using the Person correlation coefficient. The significant level was considered at *p* < 0.05.

## 3. Results

### 3.1. Basic Data of the Study Participants

Basic participants’ data are illustrated in Table 1. There were statistically significant differences (*p* < 0.05) between male and female basic data except for participants’ age and history of fracture. Most of the male participants were highly educated (87.2%), and all of them (100.0%) were employed, compared to 45.8% and 67.6% of the females, respectively. Married participants were more among females (87.8%) compared to males (60.7%). Besides, 77.3% of the male participants were urban area residents compared to 87.4% of the female participants. Obesity, personal, and family history of osteoporosis were more common in females 38.8%, 16.9%, and 25.6% compared to males 26.0%, 3.7%, and 15.3%, respectively. In addition, more than three-quarters of the male (80.2%) and female (74.7%) participants received no educational intervention about osteoporosis (Table 1).

### 3.2. Osteoporosis Knowledge, Preventive Behaviors, and Health Beliefs Differentiated by Gender

Osteoporosis knowledge, preventive behaviors, and health beliefs, differentiated by gender, are illustrated in Table 2. There were statistically significant differences between male and female participants’ osteoporosis-related knowledge, OPB, perceived susceptibility, severity, and exercise barriers (*p* ˂ 0.05). The male participants had a higher OPB of 26.70 ± 5.18 than females of 20.32 ± 5.19. However, females have a higher knowledge 10.71 ± 4.21, perceive themselves as more susceptible to osteoporosis (20.34 ± 5.39), and have higher exercise barriers (20.11 ± 6.18) compared to males 9.97 ± 3.5, 18.79 ± 5.59 and 19.20 ± 6.57, respectively. Both males and females perceived osteoporosis as severe and perceived the benefits of exercise and calcium intake and were motivated to practice healthy behaviors without significant differences between both groups (*p* ≥ 0.05) (Table 2).

### 3.3. Osteoporosis Preventive Behaviors Items Distributed by Gender

The data illustrated statistically significant differences between male and female participants in all OPB items (*p* < 0.05). Sun exposure, consumption of food-containing calcium and Vitamin D, walking half an hour daily, practicing regular physical exercise, and having a regular health check-ups were more common among males (3.26, 3.77, 3.97, 3.46, and 4.13) compared to females (2.53, 2.9, 3.04, 2.77, and 3.01), respectively. On the other hand, females exhibit more readiness to take calcium and vitamin D supplement, no smoking, and caffeinated drinks (3.91, 4.17) compared to males (2.96, 3.07), respectively. Nevertheless, males generally had a higher OPB mean score (26.70) than females (20.32) (Table 3).

### 3.4. Association between Participants’ OPB and Their Knowledge and Health Belief and Selected Demographic Variables Using Logistic Regression Analysis

Table 4 shows that gender, BMI, osteoporosis knowledge, perceived susceptibility, exercises’ perceived benefits, perceived barriers to exercises, and calcium and health motivations were associated with high OPB. A higher probability of practicing OPB was found in male participants (AOR = 9.863; 95% CI 2.157–45.102, *p* = 0.003) than in female participants. Increasing one grade in the osteoporosis-related knowledge increased the participant’s probability to practice high OPB by 1.6 times (AOR = 1.653; 95% CI = 1.363–2.004, *p* = 0.000). Regarding health beliefs, one grade increased in perceived susceptibility (AOR = 1.775; 95% CI = 1.420–2.218, *p* = 0.000), perceived benefits to exercises (AOR = 1.350; 95% CI = 1.100–1.655, *p* = 0.004), and health motivation (AOR = 1.276; 95% CI = 1.062–1.534, *p* = 0.009), thus increasing the participant’s probability to practice high OPB by 1.7, 1.3, and 1.2 times, respectively. Furthermore, a participant who had increased BMI, perceived barriers to exercises, and calcium had decreased odds (AOR 0.337; 95% CI 0.108–1.026, *p* = 0.041), (AOR 0.934; 95% CI 0.881–0.989, *p* = 0.020), and (AOR 0.072; 95% CI 0.042–0.125, *p* = 0.000), respectively, for practicing high OPB. The Nagelkerke R Square test illustrated that 35.3% of the probability of practicing high OPB could be predicted through the model.

## 4. Discussion

Knowledge about osteoporosis is one of the significant factors correlated with OPB [31]. The present study indicated satisfactory osteoporosis-related knowledge among more than two-thirds of the participants of both sexes, with a statistically significant difference in favor of women. This result may be due to women being more worried about osteoporosis because of the general concern that osteoporosis is a female disease linked to menopause, and they are more vulnerable to it. In addition, females are more exposed to health education by healthcare providers and engaged in osteoporosis awareness programs (25.3%). The higher osteoporosis-related knowledge among female participants was similar to several prior studies in Saudi Arabia [32,33]; Malaysia [31]; Canada [34], and Southern United States [35]. However, in the study by Alrashidy et al., 2021, they reported conflicting results, as it was found that males had better knowledge of osteoporosis than females. These disparities could be attributed to participants’ basic data differences, such as educational level and employment status. In the current study, 65.3% of the participants had a higher education compared to 14.8% in the study by El-Rachidi et al. Moreover, only 17.1% of the participants were not working compared to 72% in the El-Rachidi et al. study [36].

Achieving the highest possible bone mass during early life is crucial to maintaining a healthy diet and lifestyle to prevent osteoporosis. The current study found significant differences among males and females in all OPB. Sun exposure, consuming food-containing calcium and Vitamin D, walking half-an-hour daily, practicing regular physical exercise, and having a regular health check-ups were more common among male participants than females. On the other hand, females exhibit more readiness to take calcium and vitamin D supplement, no smoking, and caffeinated drinks compared to males. In general, males had a higher overall OPB mean score than females. Other studies found a similar finding. Chen et al. 2013 showed that males had higher osteoporosis protective behavior scores than females in practicing regular physical exercise [37]. In addition, the female participants were more likely to avoid harmful behaviors like smoking, alcohol, and caffeinated drink consumption than males. Moreover, Mujamammi et al., 2021, performed a cross-sectional study to assess osteoporosis-related knowledge, attitude, and practices in Saudi Arabia. They revealed that males exhibit a higher degree of adherence to a healthy lifestyle that can reduce the risk of osteoporosis compared to females [38]. Additionally, a study by Barzanji et al. showed that females were less physically active and had less exposure to sunlight than males. They further emphasized the importance of educating young people about the crucial role of sunlight exposure in sustaining adequate levels of vitamin D. They further added that it is important to educate the participants about the optimal time for sun exposure to protect their skin while reaping the benefits of sunlight [24]. It is crucial to acknowledge the impact of gender differences on protective behavior related to exercise and intake of dietary calcium and vitamin D. However, healthcare professionals should also recognize that societal norms might compel individuals of both genders to engage in activities that are detrimental to their health. In Saudi Arabia, there are gender disparities concerning lifestyle and physical activities. Females are less allowed to practice physical exercises in general places. Sun exposure to naked skin is also unacceptable due to resinous and cultural beliefs. Furthermore, the hot weather in Saudi Arabia may negatively affect attitude toward sun exposure [39].

In the current study, both male and female participants reported higher osteoporosis-related beliefs. They perceived osteoporosis as severe, perceived the benefits of exercise and calcium intake, and were motivated to practice healthy behaviors without significant differences between both sexes. Significant differences were observed only for osteoporosis perceived susceptibility and exercise perceived barriers. In several prior studies, females generally report higher perceptions of vulnerability to osteoporosis across life [34,35,40]. The literature illustrated that females are more susceptible to osteoporosis than males for many reasons; first, females have lower bone mass and smaller bone compared to males. Second, women’s bodies are more susceptible to calcium and vitamin D depletion due to recurrent pregnancy and lactation. Third, during the menopausal period, estrogen and progesterone depletion have a negative impact on bone metabolism [41,42,43]. However, osteoporosis is an important health concern for both males and females. The National Osteoporosis Foundation reports that males are more likely to die after a hip fracture and are at a higher risk of developing osteoporosis-related fractures than prostate cancer after age 50 [5]. In addition, males can develop idiopathic osteoporosis at a young age before age-related factors become apparent [44]. Therefore, it is important to prioritize educating the male population about their vulnerability and severity of osteoporosis, as well as the exercise and calcium intake benefits.

Although many studies have reported that physical activity is necessary to enhance bone structure and reduce osteoporosis prevalence, more than half of the male and female participants had higher perceived barriers to exercise. Significant gender differences were also found for exercise perceived barriers, whereas females had higher exercise barriers compared to males. A similar finding was found by other studies [45,46]. The possible reason for this could be the influence of social and cultural norms, as physical activity is not typically incorporated into the daily routine of Saudi Arabia and is instead viewed as a recreational pursuit. To encourage healthy behaviors among middle adults, it is important to address both real and perceived barriers. Women in Saudi Arabia face a variety of barriers to participating in physical activity than men, with lack of time being the most significant external barrier. Other barriers include lack of social support, low self-confidence, and limited access to facilities and resources. These factors, combined with cultural and traditional barriers, may contribute to lower levels of physical activity among women in Saudi Arabia [47,48,49].

Despite the widely recognized advantages of physical activity, motivating sedentary individuals to initiate and maintain exercise routines is a significant obstacle. Consequently, Saudi populations, especially women, should create ways to increase access to physical activity options in educational institutions, professional environments, and local communities. Since many people from both sexes in this age group were employed, employers can play a significant role by providing resources such as time and exercise. This can be achieved by offering release time during work hours, paying for gym memberships, and organizing educational interventions on healthy lifestyles [35]. Research has shown that workplace health promotion programs have a positive impact on health behaviors, biometric measures, and financial outcomes [50]. In addition, healthcare providers have a crucial responsibility to promote physical activity by regularly assessing and advising patients on increasing their physical activity levels, enhancing their fitness, and decreasing sedentary behaviors. They can also promote physical activity by creating innovative healthcare settings that encourage physical activity for patients and communities [51].

In the current study, BMI was a negative predictor for OPB. In other words, an increased BMI is a precipitating factor for decreased exercise. These results may be attributed to the sedentary lifestyle practiced by overweight and obese persons. Along the same line, Dewi et al. explored the relationship between BMI and physical fitness. They found that increased BMI is strongly correlated to decreased physical fitness [52]. Additionally, Ðošić et al. elaborated that older women with a higher BMI scored better in the environmental than physical domains of quality of life [53]. In addition, Cárdenas et al. found that high BMI is an important factor that may contribute to decreased physical activities and lead to serious health consequences [54]. Therefore, our study suggests an increased awareness regarding this major public health issue through several educational interventions that improve awareness, correct misbeliefs, and increase adherence to OPB considering gender differences within the community.

The current study shows that gender, osteoporosis knowledge, perceived susceptibility, exercises’ perceived benefits, perceived barriers to exercises, and calcium and health motivations were associated with high OPB. In the same line, Rastgoo et al. examined the predictors of OPB using the BASNEF model and reported that it was significantly associated with osteoporosis related knowledge [55]. Furthermore, Elgzar et al. found that knowledge and health beliefs related to osteoporosis were significantly associated with OPB among premenopausal women. They further added that perceived susceptibility, perceived seriousness, exercises’ perceived benefits, and health motivations are positive predictors of high OPB [16]. Finally, Blalock et al. found knowledge and attitude as positive predictors for calcium and exercise among the premenopausal [56].

### Study Strengths and Limitations

Our research has numerous strengths, such as a large sample size that provides sufficient statistical power to explore the gender differences in osteoporosis knowledge, health beliefs, and preventive behaviors. Furthermore, the study also adopted standardized tools for data collection, such as OKAT and HBMS. However, the current study has some possible drawbacks. Firstly, the results cannot be generalized to unemployed men as all male participants in the current study were employed. Secondly, the data collected was self-reported, which may lead to biases such as recall and social desirability biases.

## 5. Conclusions

This study reveals statistically significant differences between males and females regarding osteoporosis-related knowledge, preventive behaviors, perceived susceptibility, severity, and exercise perceived barriers. It is suggested that in the future, along with the questionnaire, some haemato-chemical data (calcium levels, vitamin D) and instrumental data (bone densitometry) must be reported to compare the osteoporosis perception and preventive behavior with the existence of clinical or pre-clinical disease situations. Further research is needed to explore the determinant of osteoporosis preventive behaviors among perimenopausal women. The results also suggest gender-based educational interventions to enhance OPB by addressing osteoporosis-related knowledge, perceived susceptibility, seriousness, benefits, and health motivation.

## Figures and Tables

**Table 1 nutrients-15-03658-t001:** Basic data of the study participants.

Variables	Total Sample N = 1097		Gender				X^2^/FET/*t*-Test	*p*
			Male n = (516)		Female n = (581)			
	n	%	N	%	N	%		
**Age**							1.672	0.433
20–<35	205	18.7	103	20.0	102	17.6		
35–50	856	78.0	394	76.4	462	79.5		
>50	36	3.3	19	3.7	17	2.9		
**Age (years)/Mean (SD)**	39.50 ± 7.04		39.19 ± 8.26		39.77 ± 5.71		1.36	0.173
**BMI**							51.38	0.000 *
– Underweight (BMI < 18.5)	19	1.8	14	2.7	5	0.9		
– Normal weight (BMI = 18.5 < 25)	288	26.7	183	35.7	105	18.5		
– Overweight (BMI = 25 < 30)	419	38.8	182	35.5	237	41.8		
– Obese (BMI = 30 and more)	353	32.7	133	26.0	220	38.8		
**BMI (Mean (SD))**	29.17 ± 19.95		27.00 ± 6.26		31.09 ± 6.63		3.40	0.001 *
**Residence**							19.50	0.000 *
– Urban	907	82.7%	399	77.3	508	87.4		
– Rural	190	17.3%	117	22.7	73	12.6		
**Occupational status**							201	0.000 *
– Employed	909	82.9	516	100	393	67.6		
– Not employed	188	17.1	0	0	188	32.4		
**Monthly income**							11.61	0.003 *
– Insufficient	550	50.1	284	55.0	266	45.8		
– Sufficient	280	25.5	110	21.3	170	29.3		
– Sufficient and save	267	24.3	122	23.6	145	25.0		
**Education**							255	0.000 *
– Read and write	23	2.1	14	2.7	9	1.5		
– Secondary education	358	32.6	52	10.1	306	52.7		
– High education	716	65.3	450	87.2	266	45.8		
**Marital status**							166	0.000 *
– Married	823	75.0	313	60.7	510	87.8		
– Single	234	21.3	194	37.6	40	6.9		
– Divorced	16	1.5	5	1	11	1.9		
– Widowed	24	2.2	4	0.8	20	3.4		
**Attended educational intervention about osteoporosis**							4.77	0.29 *
– Yes	249	22.7	102	19.8	147	25.3		
– No	848	77.3	4014	80.2	434	74.7		
**Personal history of osteoporosis**							49.86	0.000 *
– Yes	117	10.7	19	3.7	98	16.9		
– No	980	89.3	497	96.3	483	83.1		
**Family history of osteoporosis**							17.73	0.000 *
– Yes	228	20.8	79	15.3	149	25.6		
– No	869	79.2	437	84.7	432	74.4		
**History of bone fracture**							2.19	0.138
– Yes	302	27.5	153	29.7	149	25.6		
– No	795	72.5	363	70.3	432	74.4		

*t*: independent sample *t*-test X^2^ chi-square test FET: Fisher Exact Test * statistically significant at 0.5.

**Table 2 nutrients-15-03658-t002:** Osteoporosis knowledge, preventive behaviors, and health beliefs differentiated by gender.

Knowledge	Total Sample N = 1097		Gender				X^2^/FET/*t*-Test	*p*
			Male n = (516)		Female n = (581)			
	n	%	n	%	n	%		
**Total knowledge**							7.81	0.005 *
– Satisfactory	774	70.6	343	66.5	431	74.2		
– Unsatisfactory	323	29.4	173	33.5	150	25.8		
**Total knowledge** (Means)	10.32 ± 3.87		9.97 ± 3.5		10.71 ± 4.21		3.20	0.001 *
**Total OPB**							171	0.000 *
– Low	461	42.0	110	21.3	351	60.4		
– High	636	58.0	406	78.7	230	39.6		
**Total OPB (Mean ± SD)**	23.32 ± 6.09		26.70 ± 5.18		20.32 ± 5.19		20.31	0.000 *
**Perceived susceptibility**							30.96	0.000 *
– low	500	45.6	281	54.5	219	37.7		
– High	597	54.4	235	45.5	362	62.3		
**Perceived susceptibility (Mean ± SD)**	19.61 ± 5.54		18.79 ± 5.59		20.34 ± 5.39		4.67	0.000 *
**Perceived severity**							0.81	0.382
– low	412	37.6	201	39.0	211	36.3		
– High	685	62.4	315	61.0	370	63.7		
**Perceived severity (Mean ± SD)**	20.73 ± 5.79		20.46 ± 5.81		20.96 ± 5.76		1.4	0.159
**Exercise benefits**							1.05	0.312
– low	166	15.1	72	14.0	94	16.2		
– High	931	84.9	444	86.0	487	83.8		
**Exercise benefits (Mean ± SD)**	24.37 ± 5.15		24.59 ± 5.40		24.17 ± 4.9		1.35	0.177
**Calcium benefits**							0.170	0.680
– low	188	17.1	91	17.6	97	16.7		
– High	909	82.9	425	82.4	484	83.3		
**Calcium benefits (Mean ± SD)**	23.83 ± 4.86		23.79 ± 5.13		23.87 ± 4.62		0.272	0.785
**Perceived exercise barriers**							4.22	0.040 *
– low	485	44.2	245	47.5	240	41.3		
– High	612	55.8	271	52.5	341	58.7		
**Perceived exercise barriers (Mean ± SD)**	19.69 ± 6.38		19.20 ± 6.57		20.11 ± 6.18		2.35	0.019 *
**Perceived Calcium intake barriers**							0.55	0.459
– low	659	60.1	316	61.2	343	59.0		
– High	438	39.9	200	38.8	238	41.0		
**Perceived Calcium intake barriers (Mean ± SD)**	15.81 ± 5.40		15.70 ± 5.80		15.91 ± 5.03		0.659	0.510
**Health Motivation**							0.605	0.483
– low	202	18.4	100	19.4	102	17.6		
– High	895	81.6	416	80.6	479	82.4		
**Health motivation (Mean ± SD)**	23.52 ± 4.75		23.43 ± 4.68		23.59 ± 4.82		0.534	0.593

*t*: independent sample *t*-test X^2^ chi-square test FET: Fisher Exact Test * statistically significant at 0.5.

**Table 3 nutrients-15-03658-t003:** Osteoporosis preventive behaviors items distributed by gender.

Osteoporosis Preventive Behaviors	Cohort		Male		Female		*t*	*p*
	Mean	SD	Mean	SD	Mean	SD		
1. Exposure to the sunlight	2.87	1.21	3.26	1.24	2.53	1.08	10.35	0.000 *
2. Consumption of food-containing calcium and Vitamin D.	3.31	1.09	3.77	1.02	2.90	0.98	14.28	0.000 *
3. Taking Calcium and Vitamin D supplementation.	3.41	1.08	2.96	0.95	3.91	1.00	15.94	0.000 *
4. No smoking and caffeinated drinks	3.59	1.08	3.07	0.97	4.17	0.90	19.34	0.000 *
5. walking half an hour daily	3.48	1.07	3.97	1.01	3.04	0.93	15.72	0.000 *
6. practicing regular physical exercise	3.09	1.17	3.46	1.18	2.77	1.05	10.105	0.000 *
7. have a regular health check-up	3.54	1.10	4.13	0.921	3.01	0.96	19.68	0.000 *
**The overall mean of the Osteoporosis preventive behaviors score**	23.32	5.18	26.70	5.18	20.32	5.19	20.31	0.000 *

*t*: independent sample *t*-test * statistically significant at 0.001.

**Table 4 nutrients-15-03658-t004:** Association between participants’ OPB and their knowledge and health belief and selected demographic variables using logistic regression analysis.

**Associated Factors**	**High OPB**	
	**AOR (95% CI)**	** *p* **
Age		0.225
– 20–<35	Ref	
– 35–50	2.964 (0.781–11.246)	0.110
– >50	1.359 (0.366–5.048)	0.647
BMI	0.337 (0.108–1.026)	0.041 *
Gender		
– Female	Ref	
– Male	9.863 (2.157–45.102)	0.003 *
Education		0.611
- Read and write	Ref	
- Secondary education	0.927 (0.944–1.038)	0.642
- High education	0.931 (0.972–1.141)	0.723
Osteoporosis related knowledge	1.653 (1.363–2.004)	0.000 **
Perceived susceptibility to osteoporosis	1.775 (1.420–2.218)	0.000 **
Perceived severity of osteoporosis	1.051 (0.889–1.243)	0.560
Perceived benefits of exercise	1.350 (1.100–1.655)	0.004 *
Perceived benefits of calcium	1.045 (0.869–1.257)	0.638
Perceived Barriers to Exercise	0.934 (0.881–0.989)	0.020 *
Perceived barriers to calcium intake	0.072 (0.042–0.125)	0.000 **
Health motivation	1.276 (1.062–1.534)	0.009 *
Log likelihood (840.749)	Cox & Snell R Square (0.288)	Nagelkerke R Square (0.353)

AOR: Adjusted Odd Ratio. CI: Confidence Interval. * significant at *p* ˂ 0.05. ** significant at *p* ˂ 0.001.

## Data Availability

All data supporting this research report are available as requested from the corresponding author (W.T.E., wtelgzar@nu.edu.sa).

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
