# Peer review of "Gender Disparities in Osteoporosis Knowledge, Health Beliefs and Preventive Behaviors in Najran City, Saudi Arabia"

_nutrients, 2023, doi:10.3390/nu15163658_

Round 1

Reviewer 1 Report

Argument widely debated by the scientific community, therefore not new, but interesting for the conclusions, in fact in medicine diseases are often standardized by sex and clinical evolutions or sporadic cases with little epidemiological evidence are underestimated. scientific dissemination is the best way to prevent pathologies and/or treat them directly, avoiding complications. good study with adequate numbers of population studied, good collection of data and information, supported by the university ethics committee and also by a health ethics commission of the city in which the research was carried out. good data collection and related statistical processing. excellent bibliographic research to support methods and research.

Author Response

Reviewer 1:

Dear reviewer, many thanks for your effort and time spent in reviewing our manuscript. It is a great honor for us to benefit from your great experience.

Reviewer comment

Authors responses.

Argument widely debated by the scientific community, therefore not new, but interesting for the conclusions, in fact in medicine diseases are often standardized by sex and clinical evolutions or sporadic cases with little epidemiological evidence are underestimated. scientific dissemination is the best way to prevent pathologies and/or treat them directly, avoiding complications. good study with adequate numbers of population studied, good collection of data and information, supported by the university ethics committee and also by a health ethics commission of the city in which the research was carried out. good data collection and related statistical processing. excellent bibliographic research to support methods and research.

Many thanks for your efforts and time spent in reviewing our manuscript.

Reviewer 2 Report

Thank you for sending this article for review. Below are some comments.

Statistical P-values are compared with the best ones clearly marked in the table subject.

The novelty of this study is low. It is well known that osteoporosis is related to education level and gender.

Thank you for sending this article for review. Below are some comments.

Statistical P-values are compared with the best ones clearly marked in the table subject.

The novelty of this study is low. It is well known that osteoporosis is related to education level and gender.

Author Response

Reviewer 2:

Dear reviewer, many thanks for your effort and time spent in reviewing our manuscript. It is a great honor for us to benefit from your great experience.

Reviewer comment

Authors responses.

Statistical P-values are compared with the best ones clearly marked in the table subject.

Many thanks for your comment.

The novelty of this study is low. It is well known that osteoporosis is related to education level and gender.

Thanks for your comment, but in the current study, we did not compare osteoporosis incidence among males and females as it is well known it is more common among females, especially during the postmenopausal period. In the current study, we tried to explore if the woman increased risk for osteoporosis affected their knowledge, health beliefs, and preventive behaviors related to osteoporosis. Furthermore, when searching in literature, no studies conducted in Saudi Arabia discussed osteoporosis-related knowledge, health beliefs, and preventive behaviors from a comparative point of view among Saudi males and females. Therefore, the Argument is widely debated by the scientific community, and it is interesting for the conclusions.

Reviewer 3 Report

Thank you for the opportunity to review the manuscript “Comparing osteoporosis knowledge, health beliefs and preventive behaviors among adult male and female in Najran city Saudi Arabia”. The manuscript is well-written and the topic is really interesting. However, I have some suggestions that could increase the readability and the importance of the data.

-          The title is too long and does not underline the focus of your manuscript

Abstract

-          The mean values that you reported in the abstract are a little confusing. Maybe you can omit them.

-          You can report just two decimals for the coefficient of correlation (r)

Introduction

I have found the introduction complete and comprehensible; however, some paragraphs seem a little bit disconnected from each other resulting in not fluent reading.

Materials and methods

-          Line 92: What does it mean that “around 1131 participants were recruited”? How many participants were initially recruited and how many did you lose? Please be precise

-          I think that it would be important to add the questionnaire to the supplementary material

-          Have you, or anyone, validated this questionnaire? For example, I do not understand if the OPBS is a validated questionnaire or not

Results

The results are interesting, but you can add importance by dividing the sample into age classes (for example less and more than 50 years old) and adding a multivariate analysis.

Author Response

Reviewer 2:

Dear reviewer, many thanks for your effort and time spent in reviewing our manuscript. It is a great honor for us to benefit from your great experience. The modifications done based on your valuable comments are highlighted in yellow. 

Reviewer comment

Authors responses.

Page (line)

The title is too long and does not underline the focus of your manuscript

Done

Page1 (2.3)

Abstract

-          The mean values that you reported in the abstract are a little confusing. Maybe you can omit them.

We put the mean between breaks and omitted the stander deviation. Hoping it is clear now.

Page 1 (13,14,15)

You can report just two decimals for the coefficient of correlation (r)

Done

Page1 (17,18)

Introduction

I have found the introduction complete and comprehensible; however, some paragraphs seem a little bit disconnected from each other resulting in not fluent reading.

Done

Page 1 (35)

Page 2 (49)

Page 2 (63-67)

Materials and methods

-          Line 92: What does it mean that “around 1131 participants were recruited”? How many participants were initially recruited and how many did you lose? Please be precise

Done

Page 3 (95,96)

 I think that it would be important to add the questionnaire to the supplementary material.

Done

Supplementary file with this document

  Have you, or anyone, validated this questionnaire? For example, I do not understand if the OPBS is a validated questionnaire or not

Part 2: Osteoporosis preventive behavior scale (OPBS): developed and tested for validity and reliability by the researchers.

The OPBS was validated in the current study by a jury of four experts and a biostatistician, and the Cronbach alpha coefficient showed good reliability in the current study for the OPB scale (r=0.821).

Part 3: Osteoporosis Knowledge Assessment Tool (OKAT) developed by  Winzenberg et al., 2003. The Arabic version of OKAT had good internal consistency (r = 0.824) [29].

Part 4: Osteoporosis Health Beliefs Scale (OHBS) developed by Kim et al., 1991. OHBS Arabic version had high reliability according to Sayed-Hassan & Bashour (r = 0.806) [29]

Page 3 (112,113,114)

Page 3 (122,123)

Page 3 (130, 131)

Results

The results are interesting, but you can add importance by dividing the sample into age classes (for example less and more than 50 years old) and adding a multivariate analysis.

Done

Page 4 (171)

Page 7 (201-217)

Page 8 (table 4)

Page 10 (326-335)

Round 2

Reviewer 2 Report

After revision, it is more readable.